# Modulatory interactions between the default mode network and task positive networks in resting-state

Xin Di and Bharat B. Biswal

Department of Biomedical Engineering, New Jersey Institute of Technology, Newark, NJ, USA

## ABSTRACT

The two major brain networks, i.e., the default mode network (DMN) and the task positive network, typically reveal negative and variable connectivity in resting-state. In the present study, we examined whether the connectivity between the DMN and different components of the task positive network were modulated by other brain regions by using physiophysiological interaction (PPI) on resting-state functional magnetic resonance imaging data. Spatial independent component analysis was first conducted to identify components that represented networks of interest, including the anterior and posterior DMNs, salience, dorsal attention, left and right executive networks. PPI analysis was conducted between pairs of these networks to identify networks or regions that showed modulatory interactions with the two networks. Both network-wise and voxel-wise analyses revealed reciprocal positive modulatory interactions between the DMN, salience, and executive networks. Together with the anatomical properties of the salience network regions, the results suggest that the salience network may modulate the relationship between the DMN and executive networks. In addition, voxel-wise analysis demonstrated that the basal ganglia and thalamus positively interacted with the salience network and the dorsal attention network, and negatively interacted with the salience network and the DMN. The results demonstrated complex modulatory interactions among the DMNs and task positive networks in resting-state, and suggested that communications between these networks may be modulated by some critical brain structures such as the salience network, basal ganglia, and thalamus.

## INTRODUCTION

The human brain is intrinsically organized as different networks as generally revealed by resting-state functional magnetic resonance imaging (fMRI) (*Beckmann et al., 2005*; *Golland et al., 2008*; *Yeo et al., 2011*). Brain regions within a network generally convey relatively higher connectivity than regions from different networks (*Biswal et al., 1995*; *Cordes et al., 2000*; *Greicius et al., 2003*), thus constituting modular organizations of brain functions (*Salvador et al., 2005*; *Meunier et al., 2009*; *Doucet et al., 2011*). On the other hand, brain regions that belonged to different networks generally have weaker connectivity,

Corresponding author
Bharat B. Biswal,
bbiswal@yahoo.com

however, between network communications are considered to be critical to support complex brain functions which need to integrate resources from different brain systems (*Bullmore & Sporns, 2012*; *Cole et al., 2013*).

There are two major systems in the brain; the task positive network shows consistent activations across different tasks (*Shulman et al., 1997a*), while the default mode network (DMN) shows consistent deactivations (*Shulman et al., 1997b*). These two systems reveal moment to moment anticorrelation even when the subject isn't performing explicit tasks (*Fox et al., 2005*). The negative correlation between the DMN and the task positive network becomes stronger after adolescence (*Barber et al., 2013*; *Chai et al., 2014*), and may serve as a suppression mechanism that inhibits unwanted thoughts, thus making behavior responses more reliable (*Kelly et al., 2008*; *Spreng et al., 2010*; *Anticevic et al., 2012*; *Wen et al., 2013*). Although the original study of anticorrelation has been questioned because of global regression in data processing (*Murphy et al., 2009*), further studies have shown that the negative correlation between the DMN and the task positive network is still present without global regression (*Fox et al., 2009*; *Chai et al., 2012*), and thought to be of neuronal origins (*Keller et al., 2013*). However, the controversies of negative correlation may partially due to the fact that the connectivity between the DMN and the task positive network are highly variable (*Chang & Glover, 2010*; *Kang et al., 2011*).

The negative connectivity between the task positive network and DMN has been shown to be modulated or mediated by other networks, which may provide hints on the variability of the negative correlation. Sridharan and colleagues showed that the salience network (*Seeley et al., 2007*) activated the executive network which is part of the task positive network, and deactivated the DMN during both task performing conditions and resting-state (*Sridharan, Levitin & Menon, 2008*). In addition, Spreng and colleagues suggested that the relationship between the DMN and the dorsal attention network was mediated by regions of the frontoparietal control network (*Spreng et al., 2013*). Thus, the task positive network could be further divided into different sub-networks such as the salience network, dorsal attention network, and (left and right) executive networks, and these networks may convey complex interactions with the DMN. In the present study, we aimed to investigate whether the relationship between two networks was modulated by other networks (or regions) by using physiophysiological interaction (PPI) (*Friston et al., 1997*; *Di & Biswal, 2013a*), which might provide a novel avenue to characterize complex relationships among these networks.

Specifically, we sought to systematically investigate the modulatory interaction between the DMN and task positive networks using PPI analysis on resting-state fMRI data. Spatial independent component analysis (ICA) was first performed to identify the networks of interest, including the anterior and posterior DMNs, salience, dorsal attention, left executive, and right executive networks. PPI analysis was then performed between each two of the networks using both network-wise and voxel-wise analyses. This between-network PPI analysis was used to identify networks or regions that modulate the dynamic relationship between the two predefined networks. Based on notion that the salience network played an important role in switching of large scale brain networks

(*Sridharan, Levitin & Menon, 2008*; *Menon & Uddin, 2010*), we predicted that the salience network might show interaction effects with the DMN and executive networks.

## METHODS

### Subjects

The resting-state fMRI data was derived from the Beijing_Zang dataset of the 1000 functional connectomes project (http://fcon_1000.projects.nitrc.org/) (*Biswal et al., 2010*). This dataset originally contained 198 subjects. The first 64 subjects without large head motions were included in the current analysis (40 female/24 male). The mean age of these subjects was 21.1 years (range from 18 to 26 years of age). This study involves analyzing public available dataset, which doesn't need IRB approval. Further, we didn't use any patient identification features in this study.

### Scanning parameters

The MRI data were acquired using a SIEMENS Trio 3-Tesla scanner from Beijing Normal University. 230 resting-state functional data were acquired for each subject using TR of 2 s. The resolution of the fMRI images was $3.125 \times 3.125 \times 3$ mm$^3$ with $64 \times 64 \times 36$ voxels. T1-weighted three-dimensional magnetization-prepared rapid gradient echo (MP-RAGE) images were acquired for all the subject using the following parameters: 128 slices, TR = 2530 ms, TE = 3.39 ms, slice thickness = 1.33 mm, flip angle = 7°, inversion time = 1100 ms, FOV = $256 \times 256$ mm$^2$.

### Functional MRI data analysis

#### Preprocessing

The fMRI image preprocessing and analysis were conducted using SPM8 package (http://www.fil.ion.ucl.ac.uk/spm/) under MATLAB 7.6 environment (http://www.mathworks.com). For each subject, the first two functional images were discarded, resulting in 228 images for each subject. Firstly, the functional images were motion corrected using the realign function. The head motion estimates in any of the three translational or three rotational directions for all the subjects were less than 2 mm or 2°. Next, the functional images were linearly coregistrated to the subject's own high resolution anatomical image using the coregister function. Next, subject's anatomical images were normalized to the T1 template provided by SPM package in MNI space (Montreal Neurological Institute). Then, the normalization parameters were used to normalize all the functional images into MNI space, and the functional images were resampled into $3 \times 3 \times 3$ mm$^3$ voxels. Finally, all the functional images were smoothed using a Gaussian kernel with 8 mm full width at half maximum (FWHM).

### Spatial ICA

Spatial ICA was conducted to define networks for the PPI analysis using Group ICA of fMRI Toolbox (GIFT) (http://icatb.sourceforge.net/) (*Calhoun et al., 2001*). Twenty components were extracted. Among the 20 ICA maps (see Fig. S1), we identified the DMN and task positive network components by visually comparing the IC maps with previous

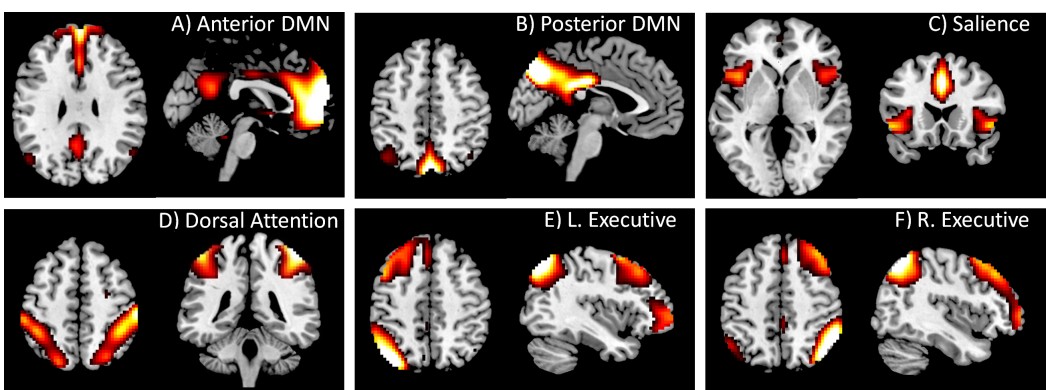

**Figure 1 DMN and task positive networks used in the PPI analysis.** These networks were defined by using spatial ICA. The IC maps were *z* transformed, and thresholded at *z* > 1.96. Maps of all 20 ICs can be found in Fig. S1.

studies (*Biswal et al., 2010*; *Cole, Smith & Beckmann, 2010*). Two components were identified as DMN, with one more anteriorly localized (Fig. 1A) and the other more posteriorly localized (Fig. 1B). We also identified four components that represented different task positive networks, i.e., the salience, dorsal attention, left executive, and right executive networks (Figs. 1C through 1F). Time series associated with these six components were obtained for each subject for following PPI analysis. To aid interpretations of the PPI results, simple correlations among the six networks were calculated for each subject. The correlation values were transformed into Fisher's *z*, and statistical significances were tested across subjects using one sample *t*-test.

## PPI analysis

Physiophysiological interaction analysis, along with its variant psychophysiological interaction, were first proposed by Friston and colleagues to characterize modulated connectivity by another region or a psychological manipulation (*Friston et al., 1997*). The present analysis focused on the modulation of connectivity by other regions or networks. Specifically, time series of two networks were used to define an interaction model using a linear regression framework.

$$y = \beta_{N1} \cdot x_{N1} + \beta_{N2} \cdot x_{N2} + \beta_{PPI} \cdot x_{N1} \cdot x_{N2} + \varepsilon$$

where $x_{N1}$ and $x_{N2}$ represented the time series of two networks. Critically, we were interested in whether the interaction term of the two time series was correlated with the time series of a given voxel or region $y$ (the effect of $\beta_{PPI}$). A positive interaction effect implies that the connectivity between the resultant region and one of the networks is positively modulated by the other network. While a negative interaction effect implies that the connectivity between the resultant region and one of the networks is negatively modulated by the other network. It should be noted that the PPI analysis is different from partial correlation analysis, which simply examines a linear relationship between two regions by controlling the activity of a third region (*Marrelec et al., 2006*). A partial

correlation is similar to the effects of $\beta_{N1}$ and $\beta_{N2}$ in the current model where the activity of $x_{N2}$ or $x_{N1}$ is controlled, respectively, which cannot directly examine the interaction of the two variables.

In practice, the time series of the two networks were deconvolved with a hemodynamic response function (hrf), so that the PPI term was calculated in the neuronal level but not hemodynamic level (*Gitelman et al., 2003*). The deconvolution procedure can in principle minimize noises when calculating PPI terms (*Gitelman et al., 2003*), and has been shown to provide better statistical results in previous empirical analysis (*Di & Biswal, 2013a*).

Before PPI analysis, the time series of each network were preprocessed in the following steps. Six rigid-body motion parameters, the first principle component time series of white matter (WM) signal, and the first principle component time series of cerebrospinal fluid (CSF) signal were regressed out from the original time series by using linear regression model. The subject specific WM and CSF masks were derived from their own segmented WM and CSF images, with a threshold of 0.99 to make sure that GM voxels were excluded from the masks. Next, a high-pass filter of 0.01 Hz was applied on the time series to minimize low frequency scanner drift. The preprocessed time series of two networks were first deconvolved with the hrf using a simple empirical Bayes procedure, so that the resulting time course represented an approximation to neural activity (*Gitelman et al., 2003*). Next, the two neural time series were detrended and point multiplied, so that the resulting time series represented the interaction of neural activity between two networks. And lastly, the interaction time series was convolved with the hrf, resulting in an interaction variable in BOLD level. The PPI terms were calculated for each pair of the six networks.

Network-wise PPI analysis was first conducted to directly examine the relationships among networks, which is similar to von Kriegstein and Giraud (*Von Kriegstein & Giraud, 2006*). In the network-wise analysis, the dependent variable was the time series of a network rather than the time series of every voxel in the brain. In the PPI linear regression model, the main effects of the two networks, and the PPI effects between them were added as independent variables along with a constant regressor. After model estimation, cross-subject one-sample $t$-tests were performed on the beta values of PPI effects. The critical $p$ value was set as $p < 0.05$ after Bonferroni correction (corresponding to a raw $p$ value of $8.33 \times 10^{-4}$ after correcting for totally 60 comparisons).

In addition, voxel-wise PPI analysis was also performed to identify regions across the whole brain that were associated with a PPI effect. PPI terms were calculated for each pair of the six networks, resulting in 15 PPI effects. Then separate PPI models were built for each subject using the general linear model (GLM) framework. The GLM model contained two regressors representing the main effects of two networks' time series, one regressor representing the PPI effect, two regressors representing WM and CSF signals, and six regressors representing head motion effects. An implicit high pass filter of 1/100 Hz was used. For each PPI effect, a 2nd-level one sample $t$-test was conducted to make group-level inference. Simple $t$ contrast of 1 or $-1$ was defined to reveal positive or negative PPI effects, respectively. The resulting clusters were first height thresholded at $p < 0.001$,

**Table 1 Mean correlations (Fisher's z scores) among the six networks.** Values in brackets represent raw *p* values of corresponding cross subject one sample *t*-test. Bold font indicates statistically significant after Bonferroni multiple comparison correction of totally 15 correlations.

| | Anterior DMN | Posterior DMN | Salience | Dorsal attention | L. executive |
|---|---|---|---|---|---|
| Posterior DMN | **0.359** **($7.01 \times 10^{-23}$)** | | | | |
| Salience | **−0.299** **($1.34 \times 10^{-16}$)** | **−0.251** **($4.75 \times 10^{-15}$)** | | | |
| Dorsal Attention | **−0.530** **($1.55 \times 10^{-28}$)** | −0.055 (0.0051) | **0.333** **($8.45 \times 10^{-16}$)** | | |
| L. Executive | **0.184** **($8.25 \times 10^{-10}$)** | **0.320** **($1.05 \times 10^{-22}$)** | 0.076 (0.0041) | 0.003 (0.87) | |
| R. Executive | **0.247** **($2.37 \times 10^{-13}$)** | **0.188** **($3.09 \times 10^{-12}$)** | **−0.142** **($1.09 \times 10^{-7}$)** | 0.004 (0.87) | **0.427** **($3.83 \times 10^{-28}$)** |

**Notes.**

L., left; R., right.

and cluster-level false discovery rate (FDR) corrected at *p* < 0.0033 based on random field theory (*Chumbley & Friston, 2009*). The cluster-level *p* value was chosen to take into account the total 15 PPI effects. The resulting clusters were labeled according to their peak coordinates using Talairach Daemon (*Lancaster et al., 2000*), after taking into account the discrepancies between MNI space and Talairach space (*Lancaster et al., 2007*).

## RESULTS

### Simple correlations among networks

The mean correlations among the six networks are listed in Table 1. There was a positive correlation between the anterior and the posterior DMNs ($M_{\text{Fisher's } z} = 0.359$). However, the correlations among the four task positive networks were mixed. The salience network revealed a positive correlation with the dorsal attention network ($M_{\text{Fisher's } z} = 0.333$), but a negative correlation with the right executive network ($M_{\text{Fisher's } z} = -0.142$). There was a positive correlation between the left and right executive networks ($M_{\text{Fisher's } z} = 0.427$). The correlations between DMN components and task positive components were also mixed. The anterior DMN showed negative correlations with the salience network ($M_{\text{Fisher's } z} = -0.299$) and the dorsal attention network ($M_{\text{Fisher's } z} = -0.530$), while positive correlations with the left executive network ($M_{\text{Fisher's } z} = 0.184$) and the right executive network ($M_{\text{Fisher's } z} = 0.247$). Similarly, the posterior DMN revealed a negative correlation with the salience network ($M_{\text{Fisher's } z} = -0.251$), while positive correlations with the left executive network ($M_{\text{Fisher's } z} = 0.320$) and the right executive network ($M_{\text{Fisher's } z} = 0.188$).

### Network-wise PPI analysis

Significant network-wise modulatory interactions are illustrated in Fig. 2 (see also Table S1 for a full list of statistics). First, positive modulatory interactions were observed among the DMNs, the salience network, and the executive networks. Positive modulatory interactions were observed among the anterior DMN, salience, and right executive networks in all of

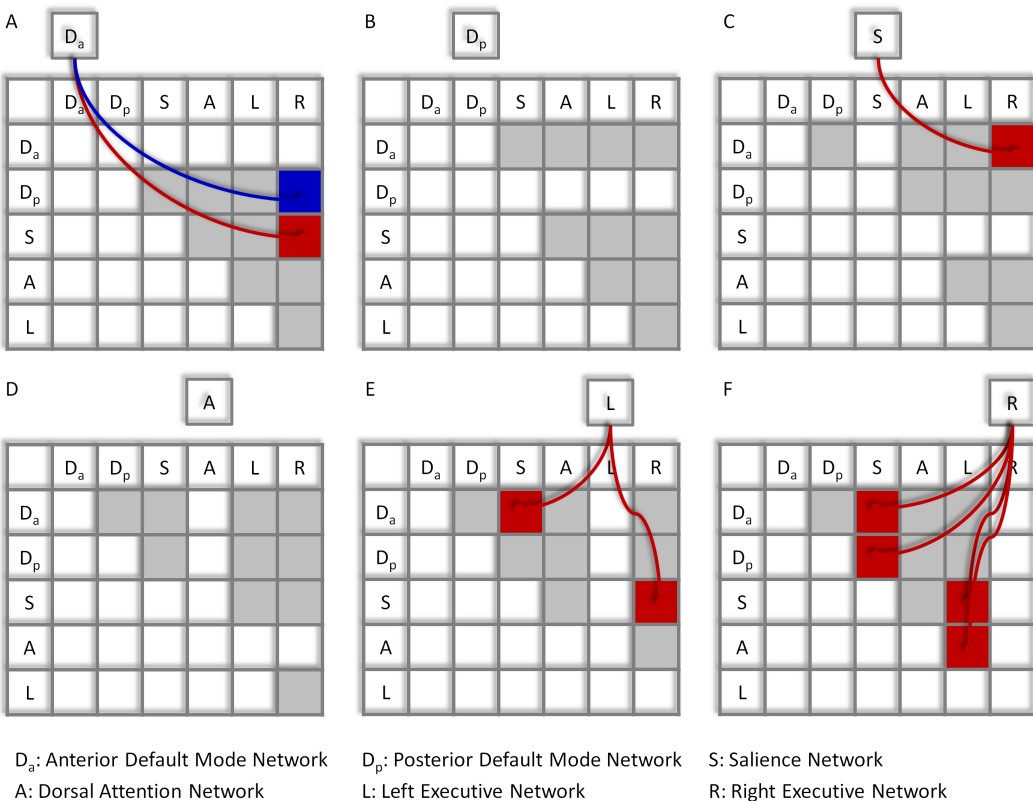

D$_a$: Anterior Default Mode Network     D$_p$: Posterior Default Mode Network     S: Salience Network
A: Dorsal Attention Network     L: Left Executive Network     R: Right Executive Network

**Figure 2 Results of the network-wise PPI analysis.** Tables indicate the PPI effects between network pairs (row vs. column). Cells outside the tables represent the dependent variables of the time series of different networks (A)–(F). Colored arrows and cells indicate significant PPI effects of a given network (outside cell) and the interaction of two ROIs (cells in the tables). Red represents positive effects, while blue represents negative effects. Cells in light gray indicate effects tested but not significant. Statistical significance was determined as $p < 0.05$ after Bonferroni correction of all 60 effects tested.

the three ways. The time series of anterior DMN were correlated with the interactions of the salience and right executive networks ($M_{beta} = 0.060; t = 4.77, p = 1.14 \times 10^{-5}$). The time series of salience network were correlated with the interactions of the anterior DMN and right executive network ($M_{beta} = 0.054; t = 4.09, p = 1.25 \times 10^{-4}$). And the time series of the right executive network were correlated with the interactions of the anterior DMN and salience network ($M_{beta} = 0.109; t = 8.27, p = 1.19 \times 10^{-11}$). The left executive time series were also correlated with the interactions of the anterior DMN and salience network ($M_{beta} = 0.048; t = 3.67, p = 4.98 \times 10^{-4}$). In addition, the time series of the right executive network were correlated with the interactions of the posterior DMN and salience network ($M_{beta} = 0.045; t = 3.81, p = 3.17 \times 10^{-4}$). Second, a negative modulatory interaction was also observed among the anterior DMN, posterior DMN and right executive network. The time series of the anterior DMN were negatively correlated with the interactions of the posterior DMN and right executive network ($M_{beta} = -0.039; t = -0.404, p = 1.48 \times 10^{-4}$). Lastly, positive modulatory interactions were also observed among task positive networks. The left executive network

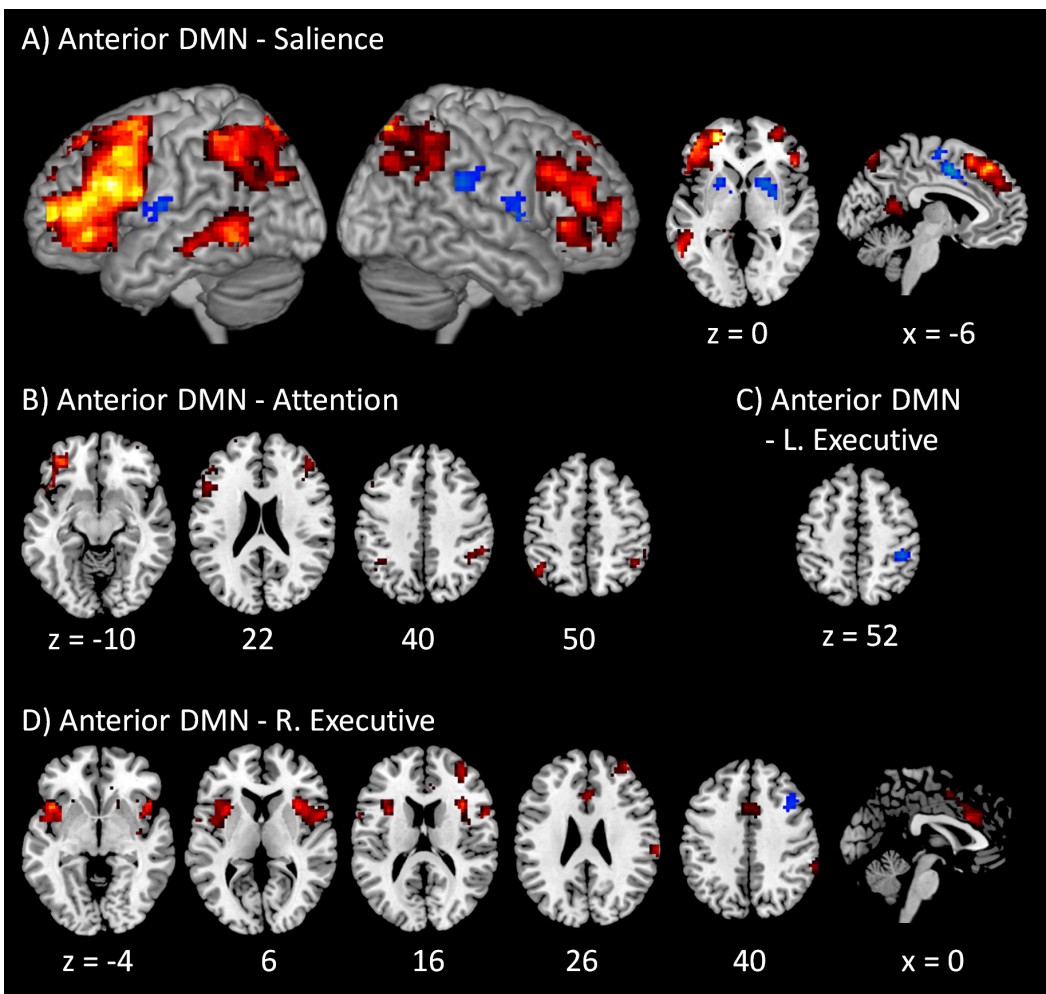

**Figure 3  Voxel-wise PPI results between the anterior DMN and task positive networks.** Clusters were thresholded at $p < 0.001$ with a cluster level FDR correction at $p < 0.0033$, which has taken into account of totally 15 voxel-wise analyses. Hot color encodes positive effects, while winter color encodes negative effects. $x$ and $z$ represent $x$ and $z$ coordinates in MNI space.

time series were correlated with the interactions of the salience and right executive network ($M_{beta} = 0.046; t = 4.01, p = 1.66 \times 10^{-4}$), and the right executive network times series were correlated with the interactions of the salience and left executive network ($M_{beta} = 0.053; t = 3.94, p = 2.06 \times 10^{-4}$). The right executive network time series were also correlated with the interaction effects of the dorsal attention and left executive networks ($M_{beta} = 0.058; t = 4.31, p = 5.91 \times 10^{-5}$).

## Voxel-wise PPI analysis

The voxel-wise PPI results of the anterior DMN with the four task positive networks are illustrated in Fig. 3. A full list of regions that showed significant PPI effects in all the fifteen voxel-wise analyses can also been found in Table S2. The regions that revealed positive modulatory interactions with the anterior DMN and salience network mainly resembled

a typical task positive network (Fig. 3A). These regions included the bilateral dorsolateral prefrontal cortex (mainly the middle and superior frontal gyrus, BA 9 and 10), bilateral parietal lobule (mainly the precuneus and inferior parietal lobule, BA 7 and 40), and left middle temporal gyrus (BA 37). Additionally, a small cluster in the posterior cingulate (BA 29) also revealed positive modulatory interactions. In contrast, several regions showed negative modulatory interactions, including the middle portion of cingulate gyrus (BA 24), bilateral putamen, and right insula (BA 13). For the modulatory interactions of the anterior DMN and dorsal attention network, positive effects were observed in the bilateral dorsolateral prefrontal cortex (mainly the middle and superior frontal gyrus, BA 9, and 47), and bilateral parietal lobule (mainly the inferior parietal lobule and supramarginal gyrus, BA 40) (Fig. 3B). No negative effects were observed. Only one region located in the right inferior parietal lobule (BA 40) revealed negative modulatory interactions with the anterior DMN and left executive network (Fig. 3C). No positive effects were observed. For the modulatory interactions of the anterior DMN and right executive network (Fig. 3D), positive effects were observed in the bilateral insula (BA 13), middle portion of cingulate gyrus (BA 24), right inferior parietal lobule (BA 40), and right middle frontal gyrus (BA 10). The bilateral insula and cingulate gyrus resembled the typical salience network. Negative effects were observed in the right middle frontal gyrus (BA 6).

The voxel-wise PPI results of the posterior DMN with the four task positive networks are shown in Fig. 4. Only positive effects were observed in the modulatory interactions of the posterior DMN and salience network, which were localized in the anterior portion of cingulate gyrus (BA 32), posterior portions of cingulate gyrus (BA 31), and left inferior parietal lobule (BA 40) (Fig. 4A). For the modulatory interactions of the posterior DMN and dorsal attention network, only positive effects were observed, which were localized in the right middle occipital gyrus (BA 19), left inferior and middle frontal gyrus (BA 44/47), right cerebellum, and left supramarginal gyrus (BA 40) (Fig. 4B). No significant modulatory interactions were found between the posterior DMN and left or right executive networks.

The PPI results of networks within the DMN and within task positive networks are shown in Fig. 5. Only negative effects were found for the modulatory interactions between the anterior and posterior DMNs, which were localized in the superior frontal gyrus (BA 6), left middle occipital gyrus (BA 19), and right precuneus (BA 7). For the modulatory interactions of the salience network and dorsal attention network (Fig. 5B), positive effects were observed in the medial frontal gyrus (BA 6), subcortical nuclei including right thalamus and left claustrum, and right postcentral gyrus (BA 2). Negative effects were observed in the left inferior frontal gyrus (BA 9). For the modulatory interactions of the salience network and left executive network (Fig. 5C), positive PPI effects were observed in the medial frontal gyrus (BA 8), left superior temporal gyrus (BA 39), and left middle frontal gyrus (BA 6). No negative effects were observed. For the modulatory interactions of the salience network and right executive network (Fig. 5D), positive effects were observed in the superior frontal gyrus (BA 8), right inferior frontal gyrus (BA 47), right superior temporal gyrus (BA 39), and right precentral gyrus (BA 9). No negative PPI

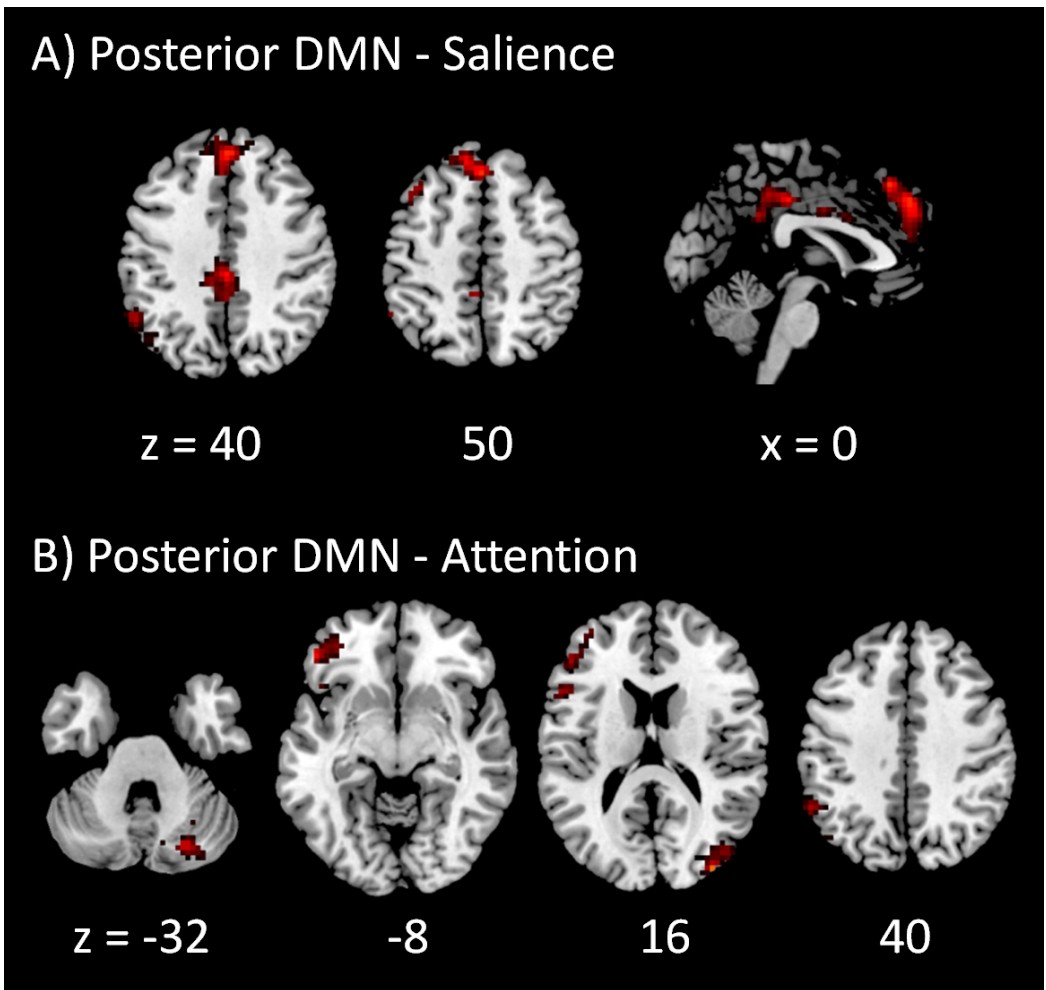

**Figure 4  Voxel-wise PPI results between the posterior DMN and task positive networks.** Clusters were thresholded at $p < 0.001$ with a cluster level FDR correction at $p < 0.0033$, which has taken into account of totally 15 voxel-wise analyses. Hot color encodes positive effects, while winter color encodes negative effects. $x$ and $z$ represents $x$ and $z$ coordinates in MNI space.

effects were observed. For the modulatory interactions of the dorsal attention network and left executive network (Fig. 5E), positive effects were observed in the left inferior parietal lobule (BA 40) and left middle frontal gyrus (BA 6). No negative effects were observed. Only one cluster in the right precuneus (BA 39) revealed positive modulatory interactions with the dorsal attention network and right executive network (Fig. 5F). Lastly, for the modulatory interactions of the left and right executive networks (Fig. 5G), positive PPI effects were observed in the bilateral precuneus (BA 7). No negative effects were observed.

## DISCUSSION

Similar to previous studies, we observed negative correlations between the DMN and some task positive networks, for example between the salience network and anterior or posterior DMNs, and between the dorsal attention network and anterior DMN. However, both

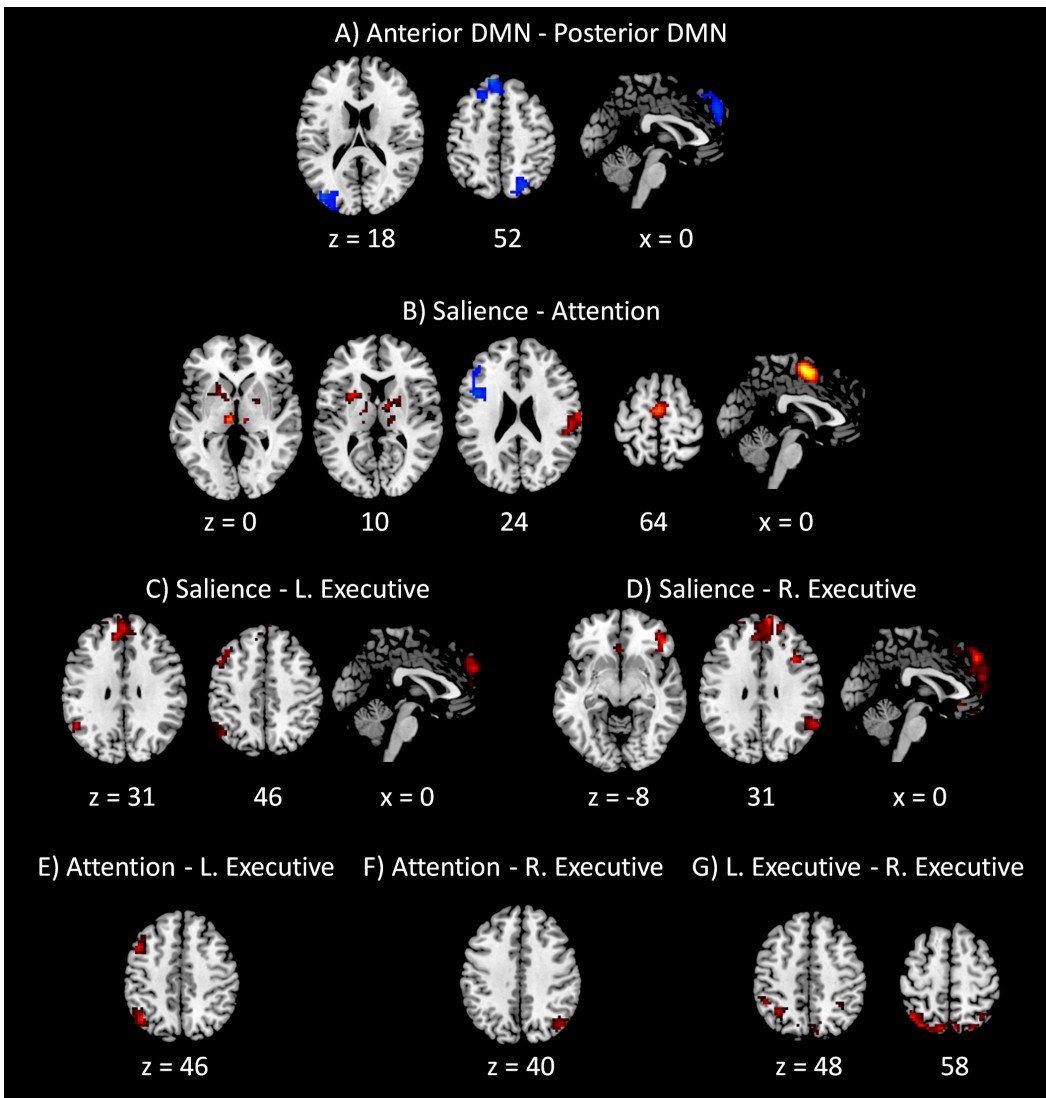

**Figure 5** **Voxel-wise PPI results between networks within the DMN and within task positive networks.** Clusters were thresholded at $p < 0.001$ with a cluster level FDR correction at $p < 0.0033$, which has taken into account of totally 15 voxel-wise analyses. Hot color encodes positive effects, while winter color encodes negative effects. $x$ and $z$ represents $x$ and $z$ coordinates in MNI space.

the anterior and posterior DMNs revealed small to moderate positive correlations with both the left and right executive networks. These results suggested complex relationships between the DMNs and different task positive networks. It should be noted that the absolute correlation values are subject to preprocessing strategies and levels of noises (*Fox et al., 2009*; *Weissenbacher et al., 2009*; *Saad et al., 2012*), so that examining the modulations of connectivity may provide complementary supports of functional interactions between networks or regions. The current PPI results can be summarized as follows. First, there were positive modulatory interactions among the DMN, the salience network and the executive networks. Second, there were negative modulatory interactions among the

anterior DMN, posterior DMN, and right executive network. Third, there were positive modulatory interactions among task positive networks, specifically the salience network with the left and right executive networks, and the dorsal attention network with the left and right executive networks. And finally, voxel-wise analysis also revealed some interesting findings, e.g., the subcortical regions such as the basal ganglia and thalamus were negatively associated with the interactions of the anterior DMN and the salience network, but were positively associated with the interactions of the salience network and the dorsal attention network.

The modulatory interaction among the DMN, the salience network, and the executive networks were mainly among the anterior DMN, the salience network and the right executive network. These results can be observed not only in the network-wise analysis, but also in the voxel-wise analysis. For example, the analysis of the anterior DMN and the salience network revealed clusters that resembled the bilateral executive network (Fig. 3A). The analysis of the anterior DMN and right executive network revealed clusters that resembled the salience network (Fig. 3D). Lastly, the analysis of the salience network and the right executive network revealed clusters that were part of the DMN (Fig. 5D). The left executive network also showed association with the interactions of the anterior DMN and the salience network in both the PPI-wise and voxel-wise analyses (Fig. 3A). In addition, the right executive network showed interactions with the posterior DMN and the salience network in the network-wise analysis. These results are consistent with our recent findings that the connectivities between the DMN regions and frontoparietal regions were positively modulated by the salience network activities, which used an independent subject sample to the current analysis (*Di & Biswal, 2013b*).

A significant PPI effect can be explained as a modulation of connectivity between two regions by the third region, or equivalently as two regions having a nonlinear multiplicative effect on the third region. Due to the nature of regression model used in PPI analysis, the role of each region can only be implied in conjunction with other evidences such as brain anatomy and causal influences. Among the DMN, salience, and executive networks, the salience network may play a critical role. Anatomically, the salience network contains a special type of neurons termed von Economo neuron (*Allman et al., 2010*), which are spindle like bipolar neurons with thick axons. These properties may enable von Economo neurons to rapidly pass information from the salience network regions to other brain regions (*Butti et al., 2009*). In terms of causal influences, studies using Granger causality analysis suggested that the salience network exerted influence to both the DMN and executive networks (*Sridharan, Levitin & Menon, 2008*; *Liao et al., 2010*; *Deshpande, Santhanam & Hu, 2011*; *Yan & He, 2011*). Taken together, a possible explanation of the PPI results may be that the salience network, in addition to activating the executive network and deactivating the DMN (*Sridharan, Levitin & Menon, 2008*), directly modulate the relationship between the executive network and DMN.

The modulation may reflect that saliency signals conveyed by the salience network increase the communication between the executive system and internal oriented system. Alternatively, because the absolute connectivity between the executive network and

the DMN is subject to preprocessing strategies, and these two networks are generally considered as anticorrelated (*Fox et al., 2005*; *Chai et al., 2012*; *Keller et al., 2013*), it is also possible that the modulation may reflect decreased anticorrelation between the DMN and executive networks. The decreased anticorrelation might suggest an absence of modulation of top-down signals from the DMN to central executive regions (*Anticevic et al., 2012*). In line with this notion, impaired salience network functions in patients of schizophrenia is coincidentally associated with altered connectivity between the executive network and DMN (*Manoliu et al., 2013*; *Manoliu et al., 2014*). The modulatory model of the salience network on the executive network and DMN may provide a novel avenue to understand dysfunctions of network communications in patients with schizophrenia (*Menon, 2011*).

In contrast, negative modulatory interactions were observed among the anterior and posterior portions of the DMN and the right executive network, which were evident in both the network-wise analysis and the voxel-wise analysis of the anterior and posterior DMNs (Fig. 5A). The voxel-wise analysis results appear similar to our previous results using the posterior cingulate gyrus (PCC) and medial prefrontal cortex (MPFC) as seed regions (*Di & Biswal, 2013a*). These results together with the above discussed results suggest complex relationships between the DMN and executive network, which differently modulated by the salience network and different parts of the DMN.

In addition to the modulatory interactions between the DMN and task positive networks, we also observed modulatory interactions among different task positive networks. These interactions were mainly among the salience network and bilateral executive networks, and among the dorsal attention network and bilateral executive networks. The frontoparietal executive network is generally identified bilaterally when using seed-based correlations and cluster analysis (*Dosenbach et al., 2007*; *Yeo et al., 2011*), however, separate left and right lateralized frontoparietal networks can be reliably identified when using ICA (*Beckmann et al., 2005*; *Biswal et al., 2010*). The current analysis revealed a moderate correlation between the left and right executive networks (mean Fisher's z 0.43), which was the largest correlation among task positive networks, suggesting that the left and right executive networks are highly functionally related. In addition, the modulatory interactions results suggested that the relationship between the left and right frontoparietal networks may be modulated by the salience network and the dorsal attention network. A previous study has suggested that the left and right lateralized executive networks may be associated with different cognitive functions, with the left executive network more associated with language cognition, and the right counterpart more related to action inhibition and pain perception (*Smith et al., 2009*). The increased connectivity between the bilateral networks may reflect the increased communication of resources from different executive systems.

Voxel-wise analysis also identified subcortical regions that revealed modulatory interactions with different networks, notably the thalamus and basal ganglia. Specifically, the bilateral putamen of the basal ganglia revealed negative modulatory interactions with the anterior DMN and salience network (Fig. 3A), while the more medial portion of the basal ganglia (mainly the globus pallidus) and the thalamus showed positive modulatory

interactions with the salience and dorsal attention networks (Fig. 5B). The basal ganglia is functionally connected to widely distributed cortical regions (*Di Martino et al., 2008*) possibly supported by different white matter fibers (*Lehéricy et al., 2004*; *Leh et al., 2007*). Models of basal ganglia functions have suggested it to be a moderator that modulate connectivity from frontal regions to posterior visual areas to support task switching and attention shifting (*Stephan et al., 2008*; *Den Ouden et al., 2010*; *Van Schouwenburg, den Ouden & Cools, 2010*). The current results extended these notions into the resting-state, suggesting a general modulating role of the basal ganglia on connectivity between brain networks. The thalamus is a critical subcortical structure that involves many functions including attention (*O'Connor et al., 2002*; *Haynes, Deichmann & Rees, 2005*). It is possible that the salience signal from the salience network enhance the connectivity from the thalamus to the dorsal attention network to allocate attention recourses to specific stimulus (*Fan et al., 2005*). Alternatively, the salience signal might modulate top-down connectivity from the dorsal attention network to the thalamus, thus facilitating attentional gating of the salient event (*McAlonan, Brown & Bowman, 2000*; *McAlonan, Cavanaugh & Wurtz, 2008*; *Fischer & Whitney, 2012*). Further studies using causal models are needed to further clarify the dynamic relationships among the thalamus, the salience network, and the dorsal attention network (*Friston, Harrison & Penny, 2003*; *Di & Biswal, 2014*).

By applying PPI technique to brain networks in resting-state, the current study demonstrated modulatory interactions among different brain systems. Compared with our previous study that examined PPI effects of two regions within the same network (*Di & Biswal, 2013a*), the current results generally revealed larger spatial extent of significant effects. One possibility is that the time series extracted from whole brain IC maps are less noisy than the time series extracted from small spherical regions of interest. Another possibility is that the time series from two regions of the same network may be highly correlated, thus the interaction is highly correlated with the main effects. Alternatively, it may reflect that different brain regions exhibit different characterizations of modulatory interactions. Some regions may dynamically connected to different regions upon task demands, while other regions may be more likely to stably connected to same regions. Charactering the spatial distributions of modulatory interactions may strengthen our understandings of brain network dynamics. For example, identifying regions that are more likely to show modulatory interactions may help to spotlight important regions that may serve as flexible hubs that dynamically control different task specific regions (*Cole et al., 2013*).

### Funding

This research was supported by a grant from NIH 5R01NS049176. The funders had no role in study design, data collection and analysis, decision to publish, or preparation of the manuscript.

## Grant Disclosures

The following grant information was disclosed by the authors:
NIH: 5R01NS049176.

## Competing Interests

The authors declare no competing commercial or financial interests.

## Author Contributions

- Xin Di conceived and designed the experiments, analyzed the data, contributed reagents/materials/analysis tools, wrote the paper, prepared figures and/or tables, reviewed drafts of the paper.
- Bharat B. Biswal conceived and designed the experiments, wrote the paper, reviewed drafts of the paper.

## Human Ethics

The following information was supplied relating to ethical approvals (i.e., approving body and any reference numbers):

This study involves analyzing a publicly available data set, which doesn't need IRB approval. Further, we didn't use any patient identification features in this study.

## Supplemental Information

Supplemental information for this article can be found online at http://dx.doi.org/10.7717/peerj.367.

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
