# Peer review of "Modulatory interactions between the default mode network and task positive networks in resting-state"

_PeerJ, doi:10.7717/peerj.367_

## Round 0.1 · original submission · Major Revisions

Both reviewers raise important and constructive criticisms that, if carefully addressed in a revised manuscript, should lead to a substantially improved paper.

·

Basic reporting

see general comments

Experimental design

see general comments

Validity of the findings

see general comments

Additional comments

this paper implements a recently developed method of physio-physiological interaction analysis for determining the statistical interaction of different intrinsic connectivity networks on one another. the authors analyze publicly available data from a small subset of the 1000 Functional Connectomes data to show specific modulatory interactions amongst the default mode network, salience network, left and right fronto-parietal networks, and the dorsal attention network. the current study is a direct extension of the work described in Di and Biswal 2013a, Modulatory Interactions of Resting-State Brain Functional Connectivity, using the same dataset and the same method. overall, the data are preprocessed using typical methods, the statistics of the PPI model and the null hypothesis testing are appropriate, and appropriate corrections for multiple comparisons are applied. the results are interesting and internally consistent. they also agree with previous work supporting the role of the salience network as a modulator of the DMN and task-positive network. the two most important outstanding issues are to 1) expand the discussion of why it is safe to interpret the sign of the PPI interaction term (positive or negative), while this is not the case for basic correlations and 2) to reconcile the pattern of within vs. between network modulations in the current study with those from the very similar Di and Biswal 2013a study (see notes in the discussion section). lastly, one thorough reading of the paper is needed to clean up tenses, plurals, and general grammar.

introduction
16-17: grammar - "thus consitute modular organizations"
18: "smaller connectivity" doesn't sound right, "weaker" might be a better choice
24: "even when one wasn't" should be rewritten to "even when the subject wasn't"
25-26: are there additional studies available supporting the emergence on DMN/task-positive anticorrelation in adolescence? this is a broad statement.
27: "unwanted noises" doesn't make sense
27: grammar - "thus make" should be "thus making"
28: "the original paper" might read better as "the original study"
41: for the reference to the frontoparietal control network, it is unclear how this relates to the task-positive network. as best i can infer, the authors define the task-positive network as dorsal attention + frontoparietal control. if this is true, it would be helpful to explicitly state that definition so the reader is clear about the definition and overlap of the different network descriptors.
47: "revenue" should be "avenue"
53: "between network" should be "between-network"

methods
general: is there a reason only 64 of the 198 available subjects were selected? there should not be any computational bottlenecks for the analyses conducted in this paper. was the desire to choose a tight age range? otherwise, it seems strange to intentionally limit the sample and only choose 33% of the available data
preprocessing: these methods are a bit sparse. it's insufficient to just say "motion corrected". what software was used - presumably the realignment program in SPM8? what was the mean rotation and translation? what was the exclusion criterion for "large head motion"? these details are important.
78: coregistered - what software was used - presumably the coregistration program in SPM8? was the coregistration linear or non-linear? were the default settings used?
99: the authors show the regression model they used to test interactions between regions. this method is similar to partial correlation. other groups have found that partial correlation is an effective model for discerning network structure in fMRI data, notably Smith et al Neuroimage 2011, Network modelling methods for FMRI. http://www.ncbi.nlm.nih.gov/pubmed/20817103. it would be helpful for the authors to explain the difference between this model and partial correlation so that readers can justify the use of the current technique vs partial correlation in their own analyses.
107: the authors used the deconvolution method for their PPI. in addition to the gitelman reference, they should reference their previous work with the exact same dataset (Di and Biswal 2013a) demonstrating the advantages of deconvolution. a short explanation of the advantages would be beneficial.
114: a high pass filter at .01 Hz was used. why was now low pass filter used. why wasn't a bandpass filter in for example .01-.08 Hz used, as is common practice? this helps reduce signal from low frequency scanner drift in addition to high frequency signal from motion and physiological nuisance sources.
128: "statistics against zero" doesn't quite sound right
133: "each pairs" should be "each pair"

results
table 1: why are the p-values so small for relatively modest Z scores, eg Z=-.299, p=1.34x10^-16?
171-173: the results mention the task-positive network but then include the middle temporal gyrus, precuneus, and posterior cingulate, which are not typically part of that network, as shown in figure 1D/E. the result should be restated to "portions of multiple networks" or some less simplified description.
174: "cigulate gyrus" is misspelled, but also too broad a descriptor. it should specify which portions of the cingulate are included - anterior, mid, or posterior?
180: "parietal lobule" is too broad. inferior parietal lobule? supramarginal and angular gyrus?
figure 4C: these results agree nicely with the findings from figure 2B
186-187: it doesn't make sense to say "cingulate gyrus" and "anterior cingulate cortex". the neuroanatomical descriptors need to be more accurately defined here.
192: "nucleus" should be "nuclei"
196: "gyrus", not "tyrus"
211: "right inferior parietal lobule" - it appears in the figure that the superior parietal lobe is also included, no?

discussion
220-222: the authors state that "the absolute values of correlations cannot be treated seriously", due to the influence of global signal regression (GSR). no GSR was applied here. are you arguing that modulatory interactions are different, and signs can be safely interpreted? the statement that "modulatory interactions are less likely to be affected by noises" isn't a sufficient explanation. interactions are simply the scalar product of the first-order timeseries, so any ambiguity in the sign of the correlation will be propagated to the interaction term. this is a critical factor for this study as it determines whether the between-network modulations are positive or negative. thus, more in-depth support is needed for the statement that interaction term signs (positive or negative) can be safely interpreted.
224-225: this sentence is almost identical to the previous sentence
239-240: the claim is tenuous that because VENs are spindle like and have thick axons, they can therefore rapidly pass information from the salience network to other regions. the connectivity targets of VENs remain largely unknown, and how their conductance varies compared to pyramidal cells is also unknown, so this statement is a stretch.
244: "sent information" is a strong statement. "exerted influence" or some other statistical phrasing seems more appropriate.
246-247: grammar - "in addition to activate"
276: another reference for these purported specializations of the left and right executive networks would be helpful
281-282: should specify that basal ganglia had negative and positive modulatory interactions with different portions of the salience network.
284: "via" is too strong a statement. it hasn't been proven that basal ganglia FC is causally related to anatomical connectivity.
291: the thalamus has many roles including attention, sensory relay, and sleep/wakefulness regulation.
302-304: it would be difficult to measure within-system modulation, given the statistical similarity of timecourse activity for regions within a network. however, the authors mention in the abstract to their very related paper (Di and Biswal 2013a) that "positive modulatory interactions were observed within regions involved in the same system". that previous statement seems to conflict the one made here. this is especially relevant because both of these papers used the same dataset and the same PPI method. thus, some attempt should be made to explain this conflict. otherwise, a reader of both papers is left to scratch their head and wonder.

figures
figure 1: it should be noted that this DMN is very anterior heavy, similar to the "anterior" DMN subportion from Damoiseaux et al 2012 (http://www.sciencedirect.com/science/article/pii/S019745801100251X figure 1a).
figure 2: this figure is very nice and valuable for summarizing the main results of the study
figure 3: this network is left lateralized, which agrees with the figure 2 results
figure 6c: this set of regions resembles the attention network, it's surprising this isn't significant in figure 2

·

Basic reporting

The manuscript has some language issues throughout; many sentences are not grammatical and verb tenses are awkward. I think the manuscript could benefit greatly by some editing for readability.

Line 47: should this be novel "avenue" instead of "revenue"?
Line 62: what does"organically" mean in this context?

Experimental design

The between-network PPI is an interesting and unusual (as far as I know) way to examine the interaction among brain networks.

The preprocessing of the resting state data appears in line with current standards.

The spatial ICA processing also seems fine, however I have questions about how the components were identified. The manuscript simply states that they were chosen "visually" and cites Cole et al., 2010. This appears to be a general review paper, which, while it makes reference to the challenges of choosing components does not as far as I can tell describe a method for reliably doing so. I think it would be nice to have some more details in the paper about how the components were actually chosen. In my experience there are often several candidate components that may look similar to a given known network and these decisions can be difficult. Did the authors consider some more objective technique for choosing components like template matching?

PPI analysis:

I understand the description of how the PPI analysis was done, but have some questions about its meaning, which I describe in the Validity section.

In Figure 2, I don't understand what the light gray boxes are. The caption states that these are "effects tested but not significant". Does that mean that the authors did not even run tests for the white boxes? If that is the case then why were certain boxes chosen to test and not others? The methods section does not describe a procedure for choosing which interactions to test.

Validity of the findings

The network-wise PPI analysis is an interesting idea, and seems to make theoretical sense on the surface, but as far as I can tell this it the first paper to employ this method and I feel like some validation of the technique would be nice. It's a little hard to conceptualize exactly what the multiplication of the two network timecourses represents on a neural level, and specifically what positive associations with this regressor mean as opposed to negative associations. Among a group of networks that all have relationships with each other, it's very difficult to understand what exactly is the significance of a positive or negative relationship of one network to the interaction term from others. What does a correlation with one of these regressors tell us about the processing of that region? Are there alternate explanations for why a brain regions should have a relationship with the interaction terms?

For instance, since there is no directionality here, its is possible that something about the relationship between the DMN and the task positive networks affects processing in the salience network, and not vice versa. In other words, a positive result in this analysis does not necessarily support the conclusion that the 3rd brain region is "modulating" the interaction between the two interacting networks.

I don't think these concerns render the results uninteresting, I just feel like as it is they are quite open to many interpretations, and at they very least these issues could be discussed in some more detail.

---

## Round 0.2 · accepted · Accept

I am glad I served as Academic Editor for this manuscript!

·

Basic reporting

No comments

Experimental design

No comments

Validity of the findings

No comments

Additional comments

I have no further issues, the authors did an excellent job addressing all of the comments. This paper helps further understanding of complex interactions between intrinsic connectivity networks and advances a useful method for examining this.